# Tetraspanin CD9 is Regulated by miR-518f-5p and Functions in Breast Cell Migration and In Vivo Tumor Growth

**DOI:** 10.3390/cancers12040795

**Published:** 2020-03-26

**Authors:** Danielle R. Bond, Richard Kahl, Joshua S. Brzozowski, Helen Jankowski, Crystal Naudin, Mamta Pariyar, Kelly A. Avery-Kiejda, Christopher J. Scarlett, Claude Boucheix, William J. Muller, Leonie K. Ashman, Murray J. Cairns, Séverine Roselli, Judith Weidenhofer

**Affiliations:** 1School of Biomedical Science and Pharmacy, The University of Newcastle and Hunter Medical Research Institute (HMRI), Newcastle, NSW 2308, Australia; Richard.Kahl@newcastle.edu.au (R.K.); Joshua.Brzozowski@newcastle.edu.au (J.S.B.); Helen.Jankowski@newcastle.edu.au (H.J.); crystal.naudin@emory.edu (C.N.); Mamta.Pariyar@uon.edu.au (M.P.); Kelly.Kiejda@newcastle.edu.au (K.A.A.-K.); leonie.ashman@newcastle.edu.au (L.K.A.); murray.cairns@newcastle.edu.au (M.J.C.); severine.roselli@newcastle.edu.au (S.R.); Judith.Weidenhofer@newcastle.edu.au (J.W.); 2Department of Pediatrics, Emory University, Atlanta, GA 30322, USA; 3School of Environmental and Life Sciences, The University of Newcastle and Hunter Medical Research Institute (HMRI), Newcastle, NSW 2308, Australia; christopher.scarlett@newcastle.edu.au; 4INSERM U935, Université Paris-Sud, Bâtiment Lavoisier, 14 Avenue Paul-Vaillant-Couturier, F-94800 Villejuif, France; claude.boucheix@inserm.fr; 5Molecular Oncology Group, Department of Biochemistry, McGill University, Montreal, QC H3A 1A3, Canada; william.muller@mcgill.ca; 6Centre for Brain and Mental Health Research, The University of Newcastle, Newcastle, NSW 2308, Australia

**Keywords:** miR-518f-5p, breast cancer, CD9, migration, regulation

## Abstract

Breast cancer is the most commonly diagnosed and the second leading cause of cancer-related mortality among women worldwide. miR-518f-5p has been shown to modulate the expression of the metastasis suppressor CD9 in prostate cancer. However, the role of miR-518f-5p and CD9 in breast cancer is unknown. Therefore, this study aimed to elucidate the role of miR-518f-5p and the mechanisms responsible for decreased CD9 expression in breast cancer, as well as the role of CD9 in de novo tumor formation and metastasis. miR-518f-5p function was assessed using migration, adhesion, and proliferation assays. miR-518f-5p was overexpressed in breast cancer cell lines that displayed significantly lower CD9 expression as well as less endogenous CD9 3′UTR activity, as assessed using qPCR and dual luciferase assays. Transfection of miR-518f-5p significantly decreased CD9 protein expression and increased breast cell migration in vitro. Cd9 deletion in the MMTV/PyMT mouse model impaired tumor growth, but had no effect on tumor initiation or metastasis. Therefore, inhibition of miR-518f-5p may restore CD9 expression and aid in the treatment of breast cancer metastasis.

## 1. Introduction

Breast cancer is a complex, heterogeneous disease that is expected to account for 30% of all new cancer cases in females in 2020, making it the most prevalent cancer among women in the United States [1]. At an estimated 15% of female cancer-related deaths in 2020, it is the second highest cause of cancer-related mortality [1]. Therefore, new and more efficacious therapies that target breast cancer progression are required to lower the mortality rates associated with breast cancer and increase patient quality of life. This requires research that increases our understanding of cancer progression and metastasis.

CD9 is considered a metastasis suppressor, typically exhibiting low protein levels in advanced prostate cancer and metastasis, leading to poor patient prognosis [2]. Studies in breast cancer have shown that patients with high CD9 expression have significantly higher overall and relapse-free survival, and those with metastatic disease with high CD9 levels respond better to therapy [3,4,5,6]. Low CD9 expression in primary breast tumors is associated with higher metastatic potential, and CD9 gene expression is commonly downregulated in lymph node metastases compared to primary breast tumors [7,8].

Whilst CD9 protein levels are commonly lower in breast cancers [9], which cells are expressing CD9 within tumors and which tumor sub-type is being assessed significantly affects the prognostic outcome of this. For example, patients with invasive luminal A subtype breast cancers had a poorer prognosis if they had high levels of CD9 in cancer cells, whereas patients with luminal B breast cancers had a good prognosis with CD9 protein expression in stromal cells [10]. Moreover, CD9 is overexpressed in osteotropic breast cancer cells and bone metastases compared to primary tumors and visceral metastases [11]. Treatment with an anti-CD9 antibody was shown to delay homing of osteotropic breast cancer cells in the bone marrow and therefore slow bone destruction in vivo [11]. In addition, CD9 knockdown suppressed metastatic capacity of MDA-MB-231 breast cancer cells in a mouse xenograft model, with delayed cell spreading and mesenchymal stromal cell invasion observed in vitro [12]. However, the majority of the literature shows a reduction in CD9 expression in breast tumors, with a more pronounced decrease with breast cancer progression, particularly in lymph node metastases [3,4,5,8,9]. These studies collectively show that high expression of CD9 may promote bone metastasis, but low expression of CD9 appears to enhance lymph node metastasis, suggesting a dual role for CD9 in breast cancer progression.

Most in vitro studies concerning CD9 functions in breast cancer have focused on its modulation of motility/migration or epithelial to mesenchymal transition. CD9 is downregulated in vitro and in vivo in epithelial to mesenchymal transition, in which CD9 expression is associated with an epithelial phenotype and good prognosis in breast cancer patients [13]. Knockdown of CD9 in breast cancer cells has been shown to increase motility [14], however, another study found delayed cell spreading and decreased motility often involving impaired integrin signaling [15]. CD9 has also been shown to play a role in cancer cell extravasation in vivo [16], and transient overexpression of CD9 on the cell surface of MDA-MB-231 breast cancer cells resulted in increased migration [17]. Therefore, CD9 appears to play a complex role in breast cancer cell functions, particularly in breast cancer cell migration and progression to metastasis.

In the last decade, there has been extensive evidence highlighting important roles for miRNAs in cancer initiation and progression to metastasis. This is mostly due to the fact that many miRNAs are differentially expressed in cancer compared to normal tissues, and thought to regulate many genes or gene pathways that are important in cancer biology [18]. We recently provided evidence that miR-518f-5p decreased the expression of the tetraspanin CD9 in prostate cancer cells, suggesting a key regulatory role of this miRNA on the function of this tetraspanin in tumor progression and invasive capacity. Interestingly, miR-518f-5p had opposing effects in prostate cancer versus non-tumorigenic prostate cells. Indeed, transfection of miR-518f-5p in prostate cancer cells led to decreased migration whereas it increased migration in non-tumorigenic cells [19]. 

There is very little known about the functional role of miR-518f-5p in any cancer, or how CD9 levels are regulated in non-tumorigenic and tumorigenic breast cells. This study investigated the role of miR-518f-5p in the regulation of CD9 and metastasis related functions in breast cell lines. Given the important, albeit contrasting, role of CD9 in breast cancer progression to metastatic disease, this study also sought to determine the effect of Cd9 ablation on breast cancer initiation, growth, and progression to metastasis in vivo. Deletion of Cd9 had no effect on tumor initiation or metastasis in the mouse mammary tumor virus-polyoma middle tumor-antigen (MMTV/PyMT) model of breast cancer, however impaired tumor growth was observed, highlighting that loss of CD9 might actually decrease tumor growth in this particular model. Moreover, the micro-RNA, miR-518f-5p was found to modulate CD9 expression and increase breast cancer cell migration in a panel of breast cell lines. Therefore, inhibition of miR-518f-5p may restore CD9 expression in breast cancers and potentially modulate cancer metastasis.

## 2. Results

### 2.1. miR-518f-5p is Predicted to be Involved in Cancer-Associated Pathways

The functional role of miR-518f-5p has been minimally investigated. We have previously shown that miR-518f-5p plays a role in migration and adhesion of non-tumorigenic and tumorigenic prostate cells. Therefore, we utilized miRNA target prediction software TargetScan [20] coupled with PANTHER [21] to provide a comprehensive analysis of functional pathways that miR-518f-5p might regulate. The most overrepresented pathways that the 2737 predicted mRNA targets of miR-518f-5p were involved in included Wnt signaling, inflammation mediated by chemokine and cytokine signaling, integrin signaling, angiogenesis, Platelet-derived growth factor (PDGF) signaling, apoptosis, and cadherin signaling (Table 1). Based on the known roles of these pathways in cancer, miR-518f-5p is likely to have a function in cancer progression.

### 2.2. miR-518f-5p Expression Correlates with Poor Overall Survival in Breast Cancer and Its Expression is Increased in Breast Cancer Cell Lines 

Given that miR-518f-5p appears to modulate cancer-associated pathways, we assessed if there was any link between miR-518f-5p expression and overall survival rates of breast cancer patients using miRpower [22]. miR-518f was identified in the expression dataset, however it was not indicated whether this was miR-518f-3p or miR-518f-5p. High expression of miR-518f was found to significantly correlate with poor overall survival rates in breast cancer patients (Figure 1A; *n* = 1061 The Cancer Genome Atlas (TCGA); *p* = 0.00055). As an initial attempt to determine if this difference in patient outcome was related to miR-518f-5p, we interrogated the expression pattern of this miR in microarray data generated from a panel of breast cancer cell lines that range in tumorigenic and invasive potential. miR-518f-5p expression was higher in the more aggressive breast cancer cell lines (MDA-MB-231, SKBR3, and T-47D) compared to the less invasive MCF7 cells and non-tumorigenic human mammary epithelial cells (HMEC) and 184A1 cells (Figure 1B).

### 2.3. miR-518f-5p Increases Breast Epithelial Cell Migration and Adhesion

We investigated whether miR-518f-5p plays a functional role in breast cancer, given its association with poor prognosis in breast cancer. Transient transfection of a miR-518f-5p mimic led to a significant increase in non-tumorigenic 184A1 breast cell migration over 24 h (Figure 2A) and in MDA-MB-231 breast cancer cell migration over 18 h (Figure 2B), compared to non-targeting control (NTC) cells. In contrast, the increase in miR-518f-5p expression had no significant effect on 184A1 or MDA-MB-231 proliferation (Figure 2C,D). Adhesion of 184A1 cells to basement membrane extract (BME) and fibronectin (FN) significantly increased following transfection of miR-518f-5p mimic (Figure 2E), compared to NTCs, however no significant effect was observed with MDA-MB-231 cell adhesion to BME or FN (Figure 2F). 

### 2.4. miR-518f-5p Decreases CD9 Protein Expression in Non-Tumorigenic Breast and Triple Negative Breast Cancer Cells

We have previously shown that miR-518f-5p decreases the expression of CD9 in prostate cancer, so we tested whether this was also the case in breast cancer. Transfection of breast cell lines with miR-518f-5p mimic resulted in a significant decrease in CD9 total protein expression in non-tumorigenic 184A1 (Figure 2G,H; 64.12% decrease, *p* = 0.0017) and triple negative MDA-MB-231 breast cancer cells (Figure 2G,H; 36.16% decrease, *p* = 0.0287) compared to NTC cells, as measured by SDS-PAGE and Western blotting.

### 2.5. CD9 Expression is Decreased in Breast Cancer Cells and Endogenous Activity Towards the CD9 3′UTR Varies Across Breast Cell Lines

To understand the likely contribution of miR-518f-5p to the regulation of CD9 and its associated functions in breast cancer it was important to analyze the endogenous levels of CD9 and relate this to the level of activity towards the *CD9* 3′UTR, the site of miRNA action. Analysis of a panel of breast cell lines showed that the majority of tumorigenic breast cells displayed low CD9 mRNA, total protein, and cell surface protein levels compared to non-tumorigenic 184A1 cells (Figure 3), with CD9 mRNA and total protein levels showing a significant positive correlation (R^2^ = 0.76, *p =* 0.02; Figure 4A). CD9 mRNA and cell surface protein, and CD9 total protein and cell surface levels did not reach significance but showed a trend towards a positive correlation (Figure 4B,C).

The degree of influence miRNAs such as miR-518f-5p have on CD9 expression through targeting its 3′UTR was assessed using a dual luciferase assay in the same panel of breast cell lines. Differential activity at the *CD9* 3’UTR was observed across the panel of non-tumorigenic and tumorigenic breast cell lines. Specifically, transfection of the *CD9* 3′UTR construct into breast cancer cell lines resulted in lower luciferase activity than that observed in non-tumorigenic 184A1 cells (Figure 4D), suggesting that breast cancer cells have an increased amount of endogenous factors, such as miRNA, that reduce CD9 expression via acting at its 3′UTR.

### 2.6. Deletion of Cd9 Impairs Tumor Growth in the MMTV/PyMT Mouse Model

While we have shown that miR-518f-5p may be responsible for the decrease in CD9 expression commonly observed in breast cancer progression, there is still a level of controversy in the literature as to the specific role of CD9 in breast cancer. As such, we also wanted to directly assess if low CD9 expression contributes to de novo tumor formation and metastasis using the well characterized MMTV/PyMT model rather than xenograft models, which are not able to fully recapitulate the in vivo environment due to the immunodeficient nature of the mice used in these studies. 

CD9 is strongly expressed and co-localizes with integrin alpha-3 throughout normal mammary luminal epithelial cells and in the PyMT breast tumor cells at different stages of tumorigenesis (Appendix A). Deletion of *Cd9* had no obvious effect on normal mammary gland development and differentiation (Appendix A) and did not affect mammary tumor onset or the number of tumors (Figure 5A,B). There was a non-significant trend towards decreased total tumor weight (*p = 0.25*) (Figure 5C), however, *Cd9* deletion did impair tumor growth of the largest tumor per mouse, with *Cd9*^−/−^ mice displaying a significantly lower tumor weight (median 0.96g vs. 1.61g, * *p* = 0.02; Figure 5D). In contrast, no significant effect on pulmonary metastasis was observed with *Cd9* deletion (Figure 6).

## 3. Discussion

The tetraspanin CD9 is considered a metastasis suppressor in many cancer types, in which low levels of CD9 protein correlate with advanced disease and poor prognosis [2]. However, very little is known about the mechanisms of regulation of CD9 expression and its dysregulation in cancers such as breast cancer. We were the first to provide evidence of a role for miR-518f-5p in modulating CD9 protein expression and migration in prostate cancer cells [19], thus implicating miRNAs in the regulation of CD9 in cancer. Here we show that miR-518f-5p also decreases CD9 expression in breast cells and causes an increase in cell migration.

There is a striking lack of information about the functional role/s of miR-518f-5p. This miRNA belongs to a family of miRNAs which are considered primate-specific and their expression is usually confined to the placenta [23]. Downregulation of miR-518f-5p expression in placental tissues is associated with severity of fetal growth restriction in pregnancy [24]. However, with the exception of our previous work in prostate cancer [19], there are no publications on the function of miR-518f-5p in cancer, or mRNA targets of miR-518f-5p. In the present study, bioinformatics analysis showed that miR-518f-5p is predicted to regulate thousands of genes that are involved in pathways that play a prominent role in cancer, in particular in breast tumorigenesis and metastasis, such as Wnt signaling [25], inflammation [26], integrin signaling [27], and angiogenesis [28]. Therefore, it was necessary to further investigate the role and importance of miR-518f-5p in breast cancer.

Transfection of a miR-518f-5p mimic into non-tumorigenic 184A1 breast cells resulted in a significant increase in migration and adhesion to BME and FN, and transfection into MDA-MB-231 breast cancer cells led to a significant increase in migration, with no change in adhesion. In both cell lines, no significant effect on cell proliferation was observed following transfection. These results show that miR-518f-5p increases migration in non-tumorigenic and tumorigenic breast cells. Our previous study on prostate cancer showed that transfection with a miR-518f-5p mimic led to a significant increase in migration of non-tumorigenic prostate cells, and a decrease in migration of prostate cancer cells. This differential effect on breast cancer cells (MDA-MB-231) and prostate cancer cells (DU145) is likely due to the myriad of experimentally unknown targets of miR-518f-5p, their expression levels in different cell types, and the functional pathways they are involved in. Further work investigating the function of miR-518f-5p in additional breast cancer cell lines is warranted to understand how relevant this miRNA is to each breast cancer subtype.

We have previously shown in prostate cells that miR-518f-5p is capable of modulating expression of the tetraspanin protein CD9 [19]. In the current study, we observed a significant decrease in CD9 protein expression in non-tumorigenic breast and breast cancer cells following transfection of miR-518f-5p. This further cements a role for miR-518f-5p in modulating CD9 expression. It is also interesting to note that the extent of CD9 knockdown by miR-518f-5p varied depending on the cell line. This may be due to differences in transfection efficiency, the endogenous expression levels of CD9 between the cell lines, alternate mechanisms leading to CD9 expression changes and/or the availability and expression of other miR-518f-5p target mRNAs. 

The increased migration of breast cell lines following transfection of miR-518f-5p, which resulted in decreased CD9 protein expression, does not align with some of the literature on CD9 functions in breast cancer cells. There are a few studies that have shown that CD9 knockdown in breast cancer cells leads to decreased motility and cell spreading [14,15]. Therefore, the effects seen with miR-518f-5p transfection on breast cell adhesion and increased migration may be due to other mRNA targets of miR-518f-5p which are currently unknown, or may be a result of disruption of interactions between CD9 and partner proteins specifically occurring in the cell lines analyzed that are involved in these pathways. No changes to in vitro cell proliferation were observed in this study, however, impaired in vivo tumor growth was observed in *Cd9*^−/−^ mice. In contrast, there is evidence to suggest that CD9 knockdown in breast cancer cell lines leads to increased cell proliferation in 3D culture, which is not observed with standard 2D cell culture [14]. Therefore, future studies involving 3D culture of miR-518f-5p transfected breast cancer cells or introduction of miR-518f-5p into the MMTV/PYMT mouse model may provide further insight and evidence for this.

CD9 protein expression is typically decreased in aggressive and advanced stage cancers. We observed that the majority of breast cancer cell lines had significantly lower CD9 mRNA, total protein, and cell surface protein levels compared to non-tumorigenic breast cells. This was expected, as the majority of the literature shows decreased CD9 expression in breast cancers, particularly associated with progression, and in lymph node metastases [3,4,5,8,9]. However, it is not clear whether the loss of CD9 expression in breast cancer is due to differences in transcriptional or post-transcriptional regulation of CD9 in breast cancer cells compared to non-tumorigenic cells. Differential regulation of CD9 through the *CD9* 3′UTR was observed across the panel of non-tumorigenic and tumorigenic breast cell lines, with some breast cancer lines showing a high level of activity towards the *CD9* 3′UTR resulting in low protein expression. This suggests that post-transcriptional regulation of CD9 by factors such as miRNA (which binds to the 3′UTR), may play an important role in breast cancer. The mechanisms responsible for the reduction in CD9 expression often observed during cancer progression are currently unknown; however there are many miRNAs that are predicted to regulate the *CD9* 3′UTR, and the RNA-binding protein, HuR, has been shown to differentially regulate CD9 mRNA stability in different breast cancer cell lines [29]. This suggests that CD9 can be regulated by RNA-binding proteins in breast cancer, but also that other forms of post-transcriptional regulation such as miRNAs, including miR-518f-5p, may also influence CD9 expression. 

Here we show for the first time that *Cd9* deletion has no significant effect on mammary tumor onset in the MMTV/PyMT mouse model but does affect tumor growth. Moreover, no significant effect was observed on pulmonary metastasis with *Cd9* ablation in this breast cancer mouse model. This was unexpected as CD9 is known to play a suppressive role in cancer progression and metastasis in other tumor types such as prostate [30], lung [31], gastric [32], and ovarian [33] cancers. The MMTV/PyMT mouse model displays pulmonary metastasis but not metastasis to other organs. This may be important with respect to CD9 functions as *Cd9*^−/−^ transgenic adenocarcinoma mouse prostate (TRAMP) prostate cancer mice displayed increased liver metastases, but no changes to lung metastases [30]. In addition, CD9 is ubiquitously expressed and its anti-cancer functions may be dependent on its expression and interactions with other proteins in exosomes [34], other cell types such as stroma [10], and other organ sites. Therefore, a mammary specific *Cd9* knockout mouse model or syngeneic engrafting of *Cd9^−/−^* PyMT tumors into *Cd9^+/+^* mice may further aid in elucidating the role of CD9 in breast cancer progression and metastasis.

## 4. Materials and Methods 

### 4.1. Cell Lines and Maintenance

Primary human breast epithelial HMEC cells (CC-2551, Lonza, Basel, Switzerland) and immortalized breast epithelial 184A1 cells (a kind gift from A/Prof. Darren Shafren, the University of Newcastle) were cultured in mammary epithelial basal medium (MEBM) supplemented with bovine pituitary extract (BPE) (0.4%), human epidermal growth factor (hEGF) (0.1%), hydrocortisone (0.1%), GA-1000 (0.1%), and insulin (0.1%) (Lonza). MCF7 (HTB-22, American Type Culture Collection (ATCC), Manassas VA, USA), T-47D (HTB-133, ATCC), MDA-MB-231 (HTB-26, ATCC), and SKBR3 (HTB-30, ATCC; a kind gift from A/Prof. Darren Shafren, the University of Newcastle) breast cancer cell lines were cultured in RPMI-1640 (GE Healthcare, Chicago, IL, USA) supplemented with 10% FBS (Sigma–Aldrich, St. Louis, MO, USA) and 2 mM L-glutamine (GE Healthcare). All cells were maintained at 37 °C with 5% CO_2_ and used within four years of purchase from ATCC or Lonza, or authenticated using the GenePrint 10 System (Promega, Madison, WI, USA) as per the manufacturer’s instructions, and DNA fragments were detected by the Australian Genome Research Facility (AGRF) (Melbourne, VIC, Australia).

### 4.2. Bioinformatic Analyses

mRNA targets of miR-518f-5p were predicted using Targetscan [20]. Genes with a cumulative weighted context++ score less than zero, which are less likely to be false positives, were subjected to pathways analysis using PANTHER [21]. This provided a list of pathways that are common to predicted mRNA targets of miR-518f-5p. miRpower [22] is a Kaplan–Meier Plotter program that was used to determine relationships between miR-518f-5p expression and overall survival, in breast cancer patients (all subtypes), using the TCGA dataset (*n* = 1061; median follow up of 25 months) with a FDR cut-off of 20%. 

### 4.3. CD9 3′UTR Dual Luciferase Reporter Assay

Cells were seeded in 96 well plates and transfected with the *CD9* 3’UTR or empty 3’UTR renilla reporter vector (SwitchGear Genomics, Menlo Park, CA, USA) and pmiR-Report firefly luciferase transfection control vector (Promega) using Lipofectamine LTX (Invitrogen, Carlsbad, CA, USA). Renilla and firefly luciferase activity were measured as previously described, using the Dual-Glo Luciferase assay system (Promega) [19]. 

### 4.4. Transient Reverse Transfection of miRNA Mimics

184A1 and MDA-MB-231 cells were transiently transfected with miRNA mimics, non-targeting control (NTC) mimic (Bioneer Pacific, Melbourne, VIC, Australia), and lipofectamine only (Mock) using lipofectamine RNAimax (Invitrogen) as previously described [19].

### 4.5. qPCR

Total RNA was extracted using TRIzol reagent (Invitrogen) according to the manufacturer’s recommendations with the addition of glycogen during precipitation at −20 °C overnight, and RNA quantity and integrity were assessed as described previously [19,35]. Total RNA was reverse transcribed to cDNA using Superscript II reverse transcriptase (Invitrogen) in accordance with the manufacturer’s recommendations and qPCR was conducted as previously described [19].

### 4.6. miRNA Microarray

Total RNA was labeled and hybridized onto Genechip miRNA 2.0 microarrays (Affymetrix, Santa Clara, CA, USA) and miRNA expression was assessed using GeneSpring GX software (Agilent, Santa Clara, CA, USA), as previously described [19]. The miRNA microarray data is publically available on NCBI GEO (GSE146477).

### 4.7. Analysis of Protein Expression

Cellular proteins were extracted, total protein concentration determined, and SDS-PAGE and Western blotting for CD9 was conducted as previously described [19]. CD9 was detected with the anti-CD9 antibody (1AA2) and actin with anti-beta-actin (AC-15, Sigma–Aldrich). Flow cytometric analysis of CD9 cell surface protein expression was conducted as previously described with CD9 antibody (1AA2) and IgG control antibody (IB5) [19].

### 4.8. Cell Proliferation Assay

Cell proliferation was indirectly assessed using the resazurin assay as described previously [19]. Briefly, cells were seeded at 1 × 10^4^ cells/well in 96-well plates and incubated overnight at 37 °C with 5% CO_2_ and resazurin (300 μM resazurin, 78 μM methylene blue, 1mM potassium hexacyanoferrate III and 1 mM potassium hexacyanoferrate II (all from Sigma–Aldrich)) added at 19 h, and fluorescence at 544/590 nm measured 5 h later using a FLUOstar-OPTIMA plate reader (BMG labtech, Offenberg, Germany). The rate of cell proliferation was determined using the number of viable cells at 24 h, 48 h, 72 h, and 96 h, with results presented as fold change of fluorescence intensity relative to 24 h fluorescence intensity (arbitrary units).

### 4.9. Cell Migration Assay

Single cell migration was assessed with 6.5 mm, 8 μm pore sized 24-well Transwells™ (Corning, Corning, NY, USA) as previously described [19]. Briefly, cells (5 × 10^5^) were loaded with Calcein-AM (8µM; AnaSpec, Fremont, CA, USA), added to the transwells, and incubated at 37 °C with 5% CO_2_ for 6–24 h. Migratory cells were detached from the insert, labeled with 8 µM Calcein-AM, and fluorescence at 490/520 nm was measured using a FLUOstar-OPTIMA plate reader.

### 4.10. Cell Adhesion Assay

Breast cells were pre-loaded with 8 µM of Calcein-AM (AnaSpec) and seeded at 2 × 10^4^ cells/well in triplicate in a black tissue culture-treated 96-well plate (Corning) as described previously [19]. Wells were pre-coated with human plasma fibronectin (FN) (Sigma–Aldrich) and basement membrane extract (BME) (Trevigen, MD, USA) overnight at 4 °C and blocked with 1% BSA before cell seeding. Following 1 h incubation at 37 °C with 5% CO_2_, calcein signal was detected using a FLUOstar-OPTIMA plate reader (490/520 nm) to obtain an initial intensity of total cell fluorescence. Non-adherent cells were then removed by 3× phosphate-buffered saline (PBS) washes and fluorescence of adherent cells was measured. Results are shown as a ratio of cell adherence.

### 4.11. Animal Breeding

Animal studies were approved by the University of Newcastle Animal Care and Ethics Committee (ref. 1024). The well-characterized FVB/N (FVB) MMTV/PyMT mouse model carrying a mouse mammary tumor virus (MMTV) promoter-driven polyoma middle T transgene was used in the tumorigenesis experiments (line MT#634) [36]. FVB *Cd9*^+/−^ mice were produced by backcrossing for 10 generations from the original C57BL/6 *Cd9* [37] and maintained as heterozygotes and inter-crossed with MMTV/PyMT mice in order to generate experimental animals. Genotyping was performed as previously described [37]. Experimental and control littermates were co-housed throughout the experiments.

### 4.12. Animal Monitoring and Tissue Collection 

Beginning at weaning (three weeks of age), female mice were palpated twice weekly for the onset of mammary tumors. For each mouse, tumor palpation was performed in each of the 10 mammary glands, in a genotype-blinded fashion. At 12–14 weeks of age, female mice were euthanized and all the tumors were dissected and weighed. At the time of dissection and after excision of the tumors, the lungs were exposed and inflated via tracheal injection of 1 ml of 10% neutral buffered formalin (NBF) in order to inflate and fix the lung lobes. Lungs were then excised and further fixed in 10% NBF for at least 24 h before paraffin embedding.

### 4.13. Whole Mount Analysis

Whole mount analysis was performed by spreading inguinal #4 mammary glands onto poly-L-lysine slides followed by overnight fixation in 10% NBF, defatting in acetone, and overnight staining in carmine alum (0.2% carmine and 0.5% aluminum sulfate) as previously described [38]. The stained glands were then dehydrated in a graded ethanol series, incubated in xylene for 1 h, and stored in methyl salicylate. 

### 4.14. Immunofluorescence Labeling 

Immunofluorescence labeling was performed on 5 μm frozen sections as previously described [38]. Primary antibodies and dilutions used for immunofluorescence labelling were rabbit anti-α3 integrin (a kind gift from Dr. Fiona Watt, Wellcome Trust Centre for Stem Cell Research, Cambridge, UK) at 1:1000 and rat anti-CD9 (BD Biosciences) at 1:1000.

### 4.15. Lung Metastasis

All the lung lobes were dissected and processed for paraffin embedding. Five micron paraffin sections were stained with hematoxylin and eosin and slides were scanned with the Aperio™digital pathology system (Aperio; Leica Biosystems, Wetzlar, Germany). The lung area per section was measured using Aperio Scanscope (Leica Biosystems) and the metastatic burden (mm^2^ of metastases/cm^2^ lung) was calculated for each animal, using the data from three sections at least 100 μm apart, as previously described [39]. 

### 4.16. Statistical Analysis

Statistical comparisons were performed using GraphPad Prism 8 software (San Diego, California, USA). Two-tailed, unpaired t-tests were used for CD9 mRNA, protein, and cell surface levels, as well as CD9 protein levels following transfection of miRNA mimics. One-way ANOVA (Dunnett’s multiple comparison testing) was used to compare *CD9* 3′UTR targeting across breast cell lines and two-way ANOVA (Dunnett’s multiple comparison testing) for analysis of cell proliferation, adhesion, and migration. For PyMT mice, Kaplan–Meier curves were generated and statistical comparisons made using the Log rank test. In vivo tumor growth and pulmonary metastasis were assessed using the Mann–Whitney test (two-tailed). All values are expressed as mean ± SEM and differences were considered significant at *p <* 0.05.

## 5. Conclusions

This study provides the first evidence that miR-518f-5p modulates migration in breast cell lines. Bioinformatics analysis showed that miR-518f-5p is predicted to regulate an array of pathways which play an important role in cancer progression and miR-518f-5p expression was shown to correlate with poor patient outcome. Overexpression of miR-518f-5p led to downregulation of CD9 protein expression in a non-tumorigenic breast cell line and a triple negative breast cancer cell line, resulting in a significant increase in cell migration. Breast cancer cell lines displayed lower CD9 mRNA, total protein, cell surface protein expression, and increased activity towards the *CD9* 3′UTR compared to non-tumorigenic breast cell lines. Furthermore, *Cd9^−/−^* PyMT breast cancer mice displayed impaired tumor growth, with no significant change to pulmonary metastasis. The results of this study collectively suggest that miR-518f-5p is capable of modulating CD9 expression and increases non-tumorigenic breast and breast cancer cell migration in vitro and that loss of CD9 in vivo affects tumorigenesis. Therefore, therapeutic inhibition of miR-518f-5p may inhibit breast cancer progression and metastasis, and thus may hold promise as a new therapy for breast cancer patients.

## Figures and Tables

**Figure 1 cancers-12-00795-f001:**
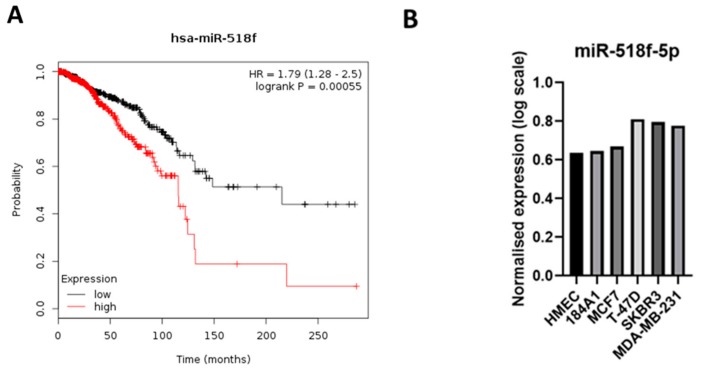
miR-518f-5p expression correlates with poor overall survival in breast cancer and its expression is increased in breast cancer cell lines. (**A**). Kaplan–Meier plot of miR-518f expression compared to the probability of overall survival of breast cancer patients (*n =* 1061 from TCGA), generated using miRpower [22]. (**B**). Micro-RNA microarray results which show miR-518f-5p expression across a panel of non-tumorigenic and tumorigenic breast cell lines (*n = 1*).

**Figure 2 cancers-12-00795-f002:**
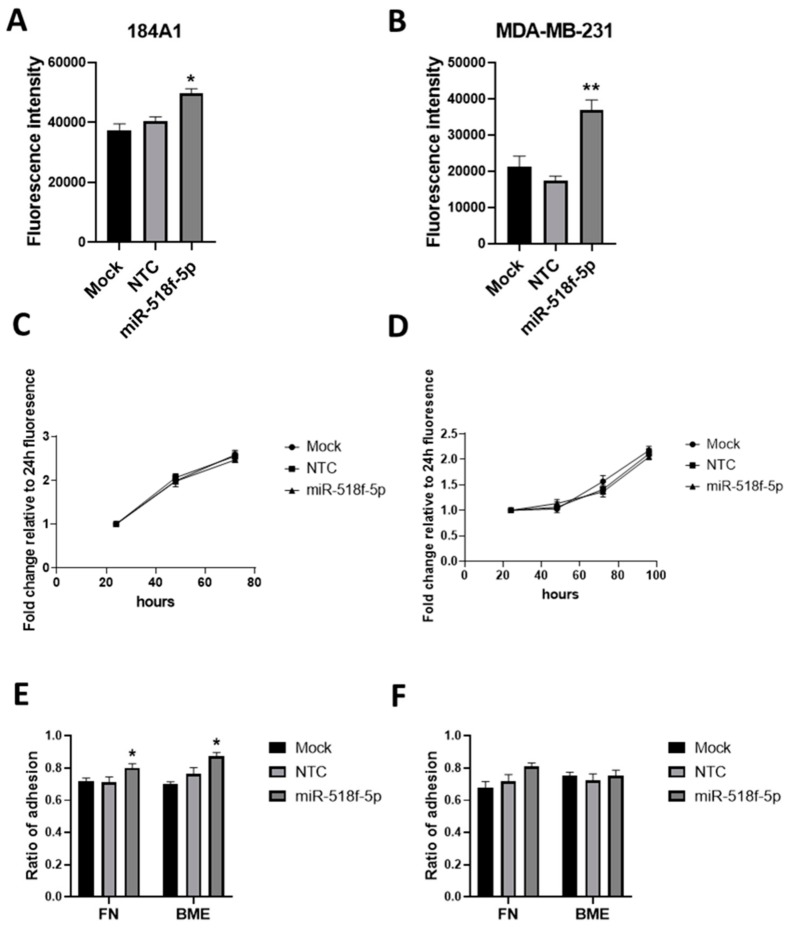
Transfection of miR-518f-5p mimic increases breast cell migration and decreases CD9 protein levels. (**A**). Cell migration in 184A1 cells following transfection of a miR-518f-5p mimic assessed using transwell migration assay. (**B**). Migration of MDA-MB-231 cells following transfection of miR-518f-5p mimic. Migration assay results are shown as fluorescence intensity (arbitrary units), * *p* = 0.02, ** *p* = 0.002. (**C**). Proliferation of 184A1 cells using the resazurin assay, after transfection of miR-518f-5p mimic. (**D**). MDA-MB-231 proliferation following transfection of a miR-518f-5p mimic, as measured using resazurin assay. Cell proliferation results are shown as fold change fluorescence relative to 24 h fluorescence (arbitrary units). (**E**). Changes in cell adhesion to basement membrane extract (BME) and fibronectin (FN) were assessed using an adhesion assay, following transfection with miR-518f-5p mimic in 184A1 cells. (**F**). Cell adhesion was assessed in MDA-MB-231 cells following transfection of miR-518f-5p mimic. Adhesion results are expressed as a ratio of adhesion (fluorescence intensity of adherent cells compared to total seeded cells, arbitrary units), FN: * *p* = 0.04, BME: * *p* = 0.01, *n =* 3. (**G**). Total CD9 protein expression was measured using SDS-PAGE and western blotting, following transfection of a miR-518f-5p mimic in 184A1 (left) and MDA-MB-231 cells (right) for 72 h. (**H**). Representative western blots of CD9 total protein expression following transfection with miR-518f-5p mimic for 72 h. Results shown as CD9 total protein normalized to beta actin protein expression, *n = 3,* * *p =* 0.04, ** *p* = 0.001. See Appendix A for full western blotting images.

**Figure 3 cancers-12-00795-f003:**
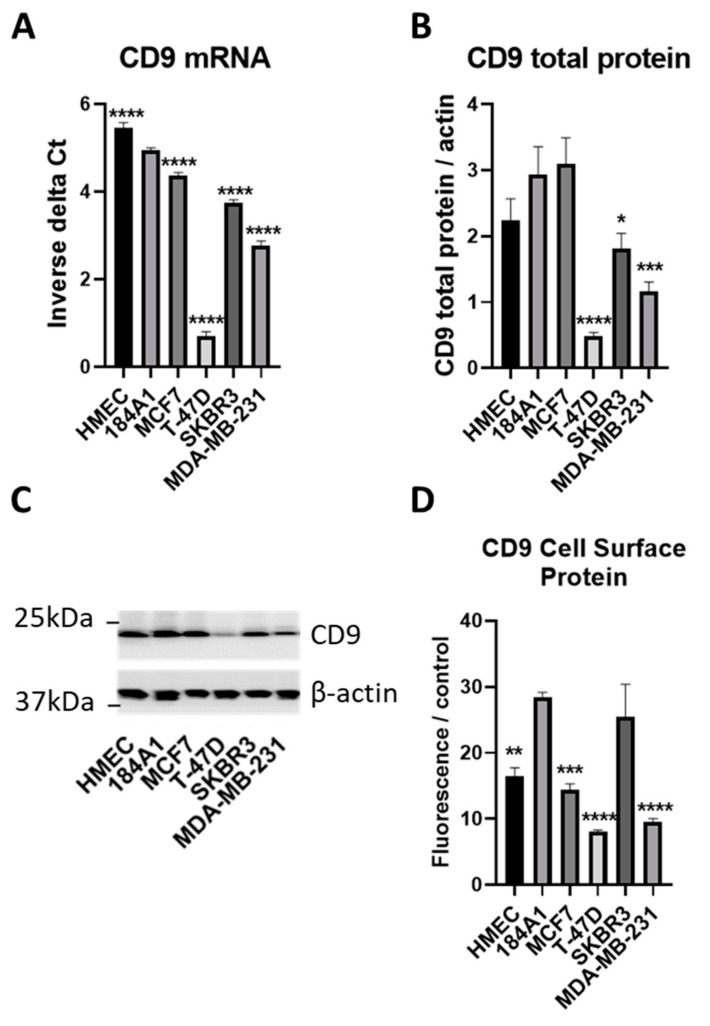
CD9 expression is decreased in non-tumorigenic and tumorigenic breast cell lines. (**A**). CD9 mRNA expression was measured using qPCR in a panel of non-tumorigenic and tumorigenic breast cell lines. Results are shown as inverse delta Ct. (**B**). CD9 total protein expression was assessed using SDS-PAGE and western blotting. Results expressed as CD9 total protein normalized to beta actin expression. (**C**). Representative western blot of CD9 total protein expression in the panel of breast cell lines. See Appendix A for full western blotting images. (**D**). Flow cytometric analysis of CD9 cell surface protein expression in breast cancer cell lines compared to non-tumorigenic breast cells. Results shown as geometric mean fluorescence intensity of CD9 antibody normalized to geometric mean fluorescence intensity of IgG antibody control. *n = 3;* * *p* = 0.04, ** *p =* 0.002, *** *p* = 0.0008, **** *p* < 0.0001.

**Figure 4 cancers-12-00795-f004:**
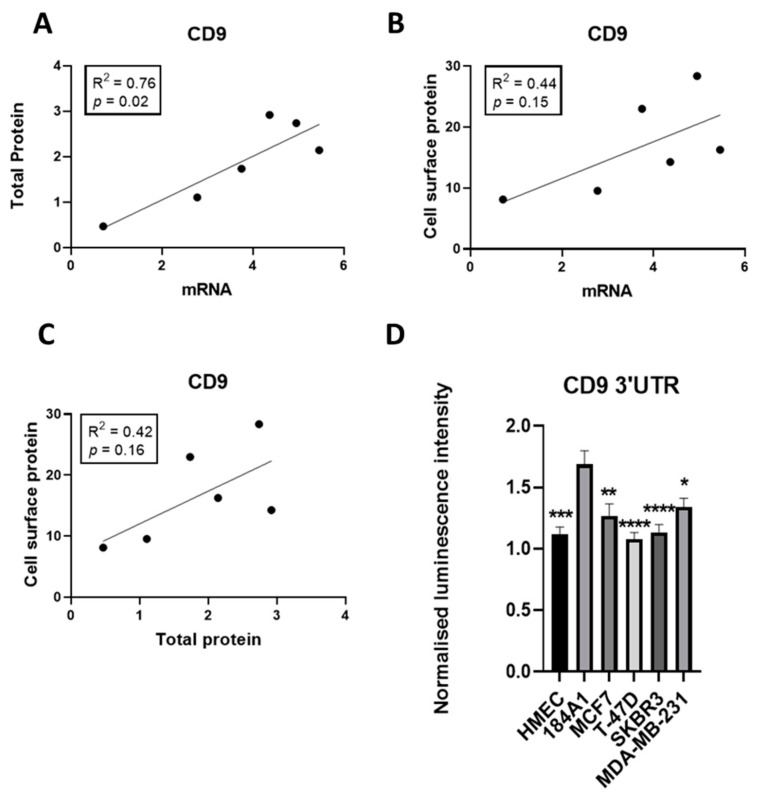
CD9 mRNA and total protein expression levels positively correlate and there is differential CD9 3′UTR activity in breast cancer cell lines. (**A**). Linear regression analysis of CD9 mRNA and total protein expression levels across the breast cell line panel. (**B**). Correlation between CD9 mRNA and CD9 cell surface protein levels as measured using linear regression analysis. (**C**). Linear regression analysis of CD9 total protein and CD9 cell surface protein expression levels. (**D**). Endogenous activity of the *CD9* 3′UTR was measured using a *CD9* 3′UTR dual luciferase reporter assay in non-tumorigenic and tumorigenic breast cell lines. *n* = 3; * *p* = 0.02, ** *p* = 0.002, *** *p* = 0.0001, **** *p* < 0.0001.

**Figure 5 cancers-12-00795-f005:**
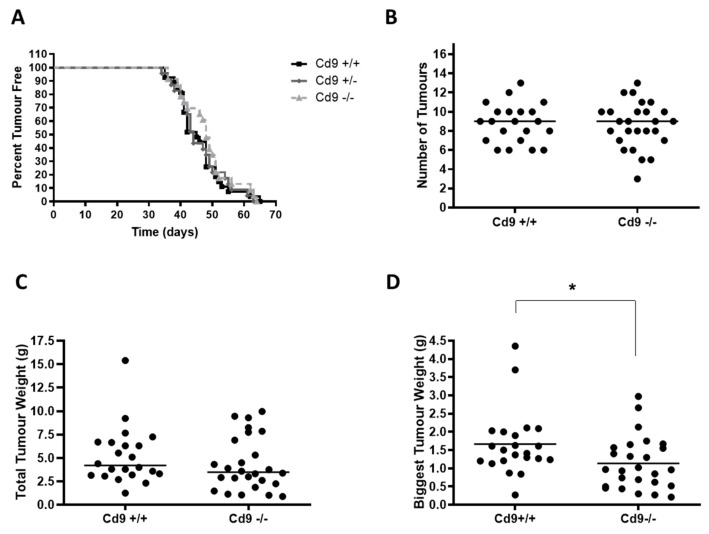
PyMT tumor growth is impaired in *Cd9*^−/−^ mice, with no change in tumor onset. (**A**). Kaplan–Meier kinetic analysis of tumor occurrence in PyMT *Cd9*^−/−^ (*n = 23)*, *Cd9*^+/−^ (*n = 23)* and *Cd9*^+/+^
*(n = 27)* mice. In order to assess the effect of *Cd9* deletion on tumor initiation, mice were palpated twice a week to detect tumor onset. (**B**). PyMT tumors were analyzed at 14 weeks of age in PyMT *Cd9*^+/+^ and *Cd9*^−/−^ mice and the total number of tumors per mouse was counted. (**C**). Total tumor weight was measured at 14 weeks of age in PyMT *Cd9^+/+^* and *Cd9*^−/−^ mice. (**D**). The weight of the biggest tumor was measured and compared between *Cd9*^−/−^ and *Cd9*^+/+^ mice; * *p* = 0.02.

**Figure 6 cancers-12-00795-f006:**
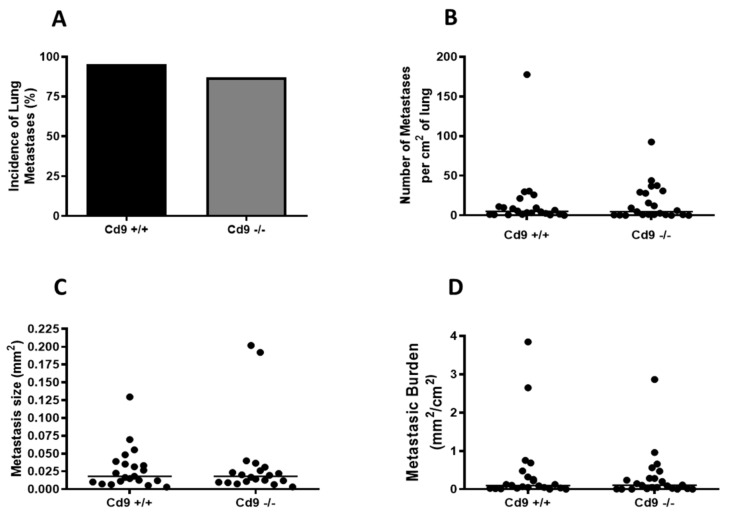
*Cd9* deletion has no significant effect on pulmonary metastasis in the MMTV/PyMT breast cancer mouse model. Pulmonary metastasis was assessed in PyMT *Cd9*^+/+^ and *Cd9*^−/−^ mice at 12–14 weeks of age. (**A**). Incidence of lung metastases in PyMT *Cd9*^−/−^ and *Cd9^+/+^* mice. (**B**). Number of metastases per mouse in PyMT *Cd9^+/+^* and *Cd9*^−/−^ mice. (**C**). Metastasis size in PyMT *Cd9^+/+^* and *Cd9*^−/−^ mice. (**D**). The metastatic burden of PyMT *Cd9^+/+^* and *Cd9*^−/−^ mice at 12–14 weeks of age.

**Table 1 cancers-12-00795-t001:** Predicted gene targets of miR-518f-5p are involved in many cancer-associated pathways.

Pathway	Number of Genes
Gonadotropin-releasing hormone receptor	51
Wnt signaling	48
Inflammation mediated by chemokine and cytokine signaling	44
Heterotrimeric G protein signaling (Gi and Gs alpha)	32
CCKR signaling	31
Integrin signaling	27
PDGF signaling	27
Angiogenesis	25
Apoptosis	25
Cadherin signaling	25
Heterotrimeric G protein signaling (Gq and Go alpha)	23
Huntington Disease	23
TGF-beta signaling	21
EGF receptor signaling	19
Cytoskeletal regulation by Rho GTPase	18
PI3 Kinase pathway	18
T cell activation	18
FGF signaling	17
p53 pathway	17
Endothelin signaling	16
B cell activation	15
Interleukin signaling	15
Ras pathway	15
Metabotropic glutamate receptor group III pathway	14
Insulin/IGF pathway—protein kinase B signaling cascade	13
Parkinson’s disease	13
Synaptic vesicle trafficking	11
Transcription regulation by bZIP transcription factor	11
Ubiquitin proteasome pathway	11
P53 pathway feedback loops 2 pathway	11
Toll receptor signaling	10

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
