# Peer review of "Tetraspanin CD9 is Regulated by miR-518f-5p and Functions in Breast Cell Migration and In Vivo Tumor Growth"

_cancers, 2020, doi:10.3390/cancers12040795_

Round 1
Reviewer 1 Report
A nice paper showing the possible connection of miR-518f-5p with CD9 expression and the prognosis of breast cancer. A clearly written paper with clear conclusions from the results obtained.
Author Response
We would like to thank the reviewers for the valuable comments to our manuscript. We have made a number of adjustments to the manuscript which have been tracked in and attached. Please refer below for specific responses to these comments.
Response: Thank you for your assessment of our paper.
Reviewer 2 Report
Comments to Manuscript Bond et al.
The aim of this work was to clarify the role of the tetraspanin CD9 in breast cancer. Here, the authors showed that CD9 expression seems to be regulated by miR-518f-5p since high miR-518f-5p were correlated with a decreased CD9 expression levels in breast cancer cells. Interestingly, lower CD9 levels were correlated with an increased migratory activity of breast cancer cell lines. Likewise, mammary tumor growth was markedly impaired in CD9 knockout mice. In brief, this is an interesting study, but, for me, the single parts of this work does not really fit together and it is unclear what is the take home message of this study. Hence, a little bit more work has to be done.
Major concerns
1) I would recommend to the authors to re-sort their data. For instance, I would start with Fig. 3 and 1A/B to show the correlation between CD9 and miR-518f-5p. Then, the data of Fig 1 C/D and Fig. 2 should be presented. Otherwise, it remains unclear why the authors started with two cell lines (mir-518f-5p overexpression experiments including CD9 expression, cell migration and cell proliferation) and then continued to work with several breast epithelial and breast cancer cell lines to present additional CD9 data.
2) To further strengthen the correlation between miR-518f-5p and CD9 and the impact on cell migration (which might play a role in breast cancer metastasis) I would recommend to the authors to perform similar experiments with a second breast cancer cell line, such as SKBR3 or T-47D.
3) Likewise, to conclude that lower CD9 expression levels might be associated with cell migration miR-518f-5p expression levels should be down-regulated with an mir-518f-5p specific antagomir. This would not only strengthen the correlation between miR-518f-5p and CD9 expression but also the authors conclusion that inhibition of miR-518f-5p may inhibit breast cancer progression and metastasis.
4) The authors used the very elegant MMTV/PyMT mouse mammary tumor model for their in vivo studies. However, it remains unclear why they have used this mouse model instead of using the cell lines that they have characterized before regarding CD9 and miR-518f-5p. For instance, MDA-MB-231 breast cancer cells are commonly used in animal studies. Hence, such breast cancer cells could be modified that they either overexpress miR-518f-5p (such cells should be more metastatic) or an appropriate miR-518f-5p antagomir or antisense vector. Of course, the MMTV/PyMT mouse mammary tumor model is much more elegant but it cannot be used to clarify the role of miR-518f-5p in breast cancer and whether it would be a putative target. So, this should be clarified.
Minor concerns
1) First page, lines 31-35: these two sentences are very similar.
2) Please label the Western Blots in the supplemental data files more clearer. Additional bands are visible between the boxes (Fig. S1) or left and right from the boxes (Fig. S2). What is seen here? A positive control?
Author Response
We would like to thank the reviewers for the valuable comments to our manuscript. We have made a number of adjustments to the manuscript which have been tracked in and attached. Please refer below for specific responses to these comments.
Reviewer 2
The aim of this work was to clarify the role of the tetraspanin CD9 in breast cancer. Here, the authors showed that CD9 expression seems to be regulated by miR-518f-5p since high miR-518f-5p were correlated with a decreased CD9 expression levels in breast cancer cells. Interestingly, lower CD9 levels were correlated with an increased migratory activity of breast cancer cell lines. Likewise, mammary tumour growth was markedly impaired in CD9 knockout mice. In brief, this is an interesting study, but, for me, the single parts of this work does not really fit together and it is unclear what is the take home message of this study. Hence, a little bit more work has to be done.
Major concerns
1) I would recommend to the authors to re-sort their data. For instance, I would start with Fig. 3 and 1A/B to show the correlation between CD9 and miR-518f-5p. Then, the data of Fig 1 C/D and Fig. 2 should be presented. Otherwise, it remains unclear why the authors started with two cell lines (mir-518f-5p overexpression experiments including CD9 expression, cell migration and cell proliferation) and then continued to work with several breast epithelial and breast cancer cell lines to present additional CD9 data.
Response: Thank you for this suggestion. We have examined several possible ways to present this data and have made a few rearrangements (although not exactly as suggested). Part C & D from figure 1 has been moved to figure 2 (as G & H) and therefore section 2.4 has been brought ahead of 2.4 and hence now referred to as 2.3. We have also added some additional text that hopefully provides some more context to the experiments and their interpretation. Briefly, the previous work we had done in prostate cancer showed a role for miR-518f-5p in metastasis related functions and the regulation of CD9. Hence, we originally investigated the interaction between the miR and CD9 in breast cancer using two common cell lines. We then progressed to understand the endogenous relevance of this by using more cell lines. Rather than continuing to add large amounts of miR-518f-5p mimic into different lines, we felt it was more appropriate to get an understanding of the endogenous activity that is present towards the 3’UTR of CD9. We feel that the adjustments we have made highlight this more than the previous version and thank you for bringing this to our attention.
2) To further strengthen the correlation between miR-518f-5p and CD9 and the impact on cell migration (which might play a role in breast cancer metastasis) I would recommend to the authors to perform similar experiments with a second breast cancer cell line, such as SKBR3 or T-47D.
Response: We appreciate that this would greatly strengthen this manuscript, however we are not in a position to conduct further experimental work of this nature at this time. As such we have included a statement within the discussion that says ‘Further work investigating the function of miR-518f-5p in additional breast cancer cell lines is warranted to understand how relevant this miRNA is to each breast cancer subtype.’
3) Likewise, to conclude that lower CD9 expression levels might be associated with cell migration miR-518f-5p expression levels should be down-regulated with a miR-518f-5p specific antagomir. This would not only strengthen the correlation between miR-518f-5p and CD9 expression but also the authors conclusion that inhibition of miR-518f-5p may inhibit breast cancer progression and metastasis.
Response: Whilst we agree that this experiment would strengthen our conclusions, again we are not in a position to undertake this experiment. The conclusion we have provided is stated as a possible mechanism rather than a definitive one in light of this lack of evidence.
4) The authors used the very elegant MMTV/PyMT mouse mammary tumor model for their in vivo studies. However, it remains unclear why they have used this mouse model instead of using the cell lines that they have characterized before regarding CD9 and miR-518f-5p. For instance, MDA-MB-231 breast cancer cells are commonly used in animal studies. Hence, such breast cancer cells could be modified that they either overexpress miR-518f-5p (such cells should be more metastatic) or an appropriate miR-518f-5p antagomir or antisense vector. Of course, the MMTV/PyMT mouse mammary tumor model is much more elegant but it cannot be used to clarify the role of miR-518f-5p in breast cancer and whether it would be a putative target. So, this should be clarified.
Response: We are sorry that the intended purpose of these experiments was not clearly stated. The intention here was to understand how much of the in vitro effects observed with miR-518f-5p were due to the identified effect of decreasing CD9 or the potential effects on the multiple pathways that the miRNA is predicted to regulate. In addition, we sought to understand if CD9 is a major player in breast cancer tumourigenesis as this was not yet known. Whilst we could have utilised xenograft models, by their nature they are not a faithful recapitulation of tumourigenesis, in part due to the required immunodeficient nature of the host. As such we took the approach to analyse the function of CD9 in the MMTV/PyMT model. Indeed and not unexpectedly the minimal effects observed in tumourigenesis in this model point to the in vitro effects of miR-518f-5p being only in part due to its effects on CD9. We have made adjustments to the wording of this section of the manuscript to explain this approach.
Minor concerns
1) First page, lines 31-35: these two sentences are very similar.
Response: We have removed the second sentence that was duplicative.
2) Please label the Western Blots in the supplemental data files more clearer. Additional bands are visible between the boxes (Fig. S1) or left and right from the boxes (Fig. S2). What is seen here? A positive control?
Response: We have added additional labelling to these figures to clarify what these bands are. In Fig S1 the samples in the other lanes were: Lanes 4, 8 and 12 contain lysates from miR-4289 mimic transfected cell lines that served as a positive control. In Fig S2 the additional samples to the indicated bands were Lanes 1, 2, 21 & 22 contain the positive control RWPE-1 protein lysate.
Reviewer 3 Report
Dear Authors
The manuscript entitled "Tetraspanin CD9 is regulated by miR-518f-5p and functions in breast cell migration and in vivo tumour growth” it is well designed in the experimental line. Also, highlights how non-coding RNAs can represent a future therapeutic target. It is my opinion that the manuscript needs a revision of grammar.
Author Response
We would like to thank the reviewers for the valuable comments to our manuscript. We have made a number of adjustments to the manuscript which have been tracked in and attached. Please refer below for specific responses to these comments.
Reviewer 3: The manuscript entitled "Tetraspanin CD9 is regulated by miR-518f-5p and functions in breast cell migration and in vivo tumour growth” it is well designed in the experimental line. Also, highlights how non-coding RNAs can represent a future therapeutic target. It is my opinion that the manuscript needs a revision of grammar.
Response: Thank you for your consideration of our manuscript and feedback. We have revised the grammar throughout and used the track changes function to highlight these adjustments.
Round 2
Reviewer 2 Report
All my concerns were fully addressed by the authors. I do not have any further comments.